# Regression of Lung Cancer in Mice by Intranasal Administration of SARS-CoV-2 Spike S1

**DOI:** 10.3390/cancers14225648

**Published:** 2022-11-17

**Authors:** Monica Sheinin, Brian Jeong, Ramesh K. Paidi, Kalipada Pahan

**Affiliations:** 1Department of Neurological Sciences, Rush University Medical Center, Chicago, IL 60612, USA; 2Division of Research and Development, Jesse Brown Veterans Affairs Medical Center, Chicago, IL 60612, USA

**Keywords:** human lung cancer cells, NNK mouse model of lung cancer, SARS-CoV-2 spike S1, ACE2, apoptosis

## Abstract

**Simple Summary:**

Lung cancer is the leading cause of cancer related deaths worldwide, with a relatively low 5-year survival rate. Although there are many therapies against lung cancer, significant improvements in overall survival have not been reported. Therefore, new effective treatment options are needed. Angiotensin-converting enzyme 2 (ACE2) is present in lungs and it has been shown that stimulation of ACE2 may be an important mechanism to control lung cancer growth. Recently during the COVID-19 pandemic, we have seen that SARS-CoV-2 binds to its receptor ACE2 via spike S1 to enter into the cells. Interestingly, we found that recombinant SARS-CoV-2 spike S1 employed its interaction with ACE2 to induce the death of human lung cancer cells and that intranasal administration of recombinant spike S1 led to regression of tumor in vivo in the lung of NNK-intoxicated mice. Therefore, intranasal administration of SARS-CoV-2 spike S1 may be beneficial for lung cancer patients.

**Abstract:**

This study underlines the importance of SARS-CoV-2 spike S1 in prompting death in cultured non-small cell lung cancer (NSCLC) cells and in vivo in lung tumors in mice. Interestingly, we found that recombinant spike S1 treatment at very low doses led to death of human A549 NSCLC cells. On the other hand, boiled recombinant SARS-CoV-2 spike S1 remained unable to induce death, suggesting that the induction of cell death in A549 cells was due to native SARS-CoV-2 spike S1 protein. SARS-CoV-2 spike S1-induced A549 cell death was also inhibited by neutralizing antibodies against spike S1 and ACE2. Moreover, our newly designed wild type ACE2-interacting domain of SARS-CoV-2 (wtAIDS), but not mAIDS, peptide also attenuated SARS-CoV-2 spike S1-induced cell death, suggesting that SARS-CoV-2 spike S1-induced death in A549 NSCLC cells depends on its interaction with ACE2 receptor. Similarly, recombinant spike S1 treatment also led to death of human H1299 and H358 NSCLC cells. Finally, 4-(methylnitrosamino)-1-(3-pyridyl)-1-butanone (NNK) intoxication led to the formation tumors in lungs of A/J mice and alternate day intranasal treatment with low dose of recombinant SARS-CoV-2 spike S1 from 22-weeks of NNK insult (late stage) induced apoptosis and tumor regression in the lungs. These studies indicate that SARS-CoV-2 spike S1 may have implications for lung cancer treatment.

## 1. Introduction

According to the Centers for Disease Control and Prevention, cancer is the number two cause of death in USA, second only to heart disease [1]. Ironically, among all cancers, lung cancer is by far the leading cause of cancer death in which more than 130,000 people die each year in USA. In fact, more people die of lung cancer than of colon, breast and prostate cancers together, approximately amounting to 25% of all cancer deaths in USA [2]. The five-year survival rate for lung cancer patients (22%) is also significantly lower than other cancers [3]. Therefore, understanding molecular mechanisms and developing an effective therapeutic approach for lung cancer are of paramount importance. Interestingly, the renin-angiotensin system plays an important role in lung tumor progression or metastasis [4]. It has been shown that the low expression of angiotensin-converting enzyme 2 (ACE2) is associated with tumor grade in lung cancer and that upregulation of ACE2 suppresses the progression of non-small cell lung cancer (NSCLC) [5,6]. For example, according to Cheng et al. [7], ACE2 overexpression could prevent acquired platinum resistance-induced tumor angiogenesis in NSCLC. Again, Feng et al. [8] have shown that overexpression of ACE2 can cause antitumor effects through suppression of angiogenesis and tumor cell invasion. Alternatively, use of ACE2 inhibitors has been found to be associated with increased risk of lung cancer in a population-based cohort study from UK [9]. Therefore, stimulation of ACE2 may be an important mechanism to control lung cancer growth.

Incidentally, due to the recent COVID-19 pandemic, we have been given a unique molecule to activate and transduce the ACE2 signaling pathway [10,11,12,13]. SARS-CoV-2 binds to its receptor ACE2 via spike S1 to enter and infect human cells [11,12,13,14]. We employed recombinant SARS-CoV-2 Spike S1 to induce apoptosis in lung cancer cells and found apoptosis and cell death in different human lung cancer cells. SARS-CoV-2 spike S1-induced lung cancer cell death was dependent on native spike S1 protein. Moreover, lung cancer cells expressed ACE2 and spike S1-mediated cancer cell death relied on the interaction of spike S1 with ACE2. To investigate the effect of spike S1 on lung tumor in vivo in mice, we used the 4-(methylnitrosamino)-1-(3-pyridyl)-1-butanone (NNK) mouse model [15]. While untreated and saline-treated NNK-intoxicated mice exhibited widespread tumor formation in the lungs, intranasal treatment of recombinant SARS-CoV-2 spike S1 from the late stage (22 weeks of NNK insult) of disease led to apoptosis and regression of lung tumor. Recombinant SARS-CoV-2 spike S1 is free of virus and does not contain any infectious component. Therefore, our results suggest that intranasal administration of recombinant SARS-CoV-2 Spike S1 may be an effective therapeutic option for lung cancer patients.

## 2. Materials and Methods

### 2.1. Reagents

Recombinant SARS-CoV-2 spike S1 (14-685) was provided by Abeomics, San Diego, CA, USA. Recombinant human ACE2 protein (18-739) was supplied by MyBiosource, San Diego, CA, USA. Human lung carcinoma cell lines (A549, H1299 and H358) and F-12K medium were obtained from ATCC, Manassas, VA. Hank’s balanced salt solution, RPMI-1640, penicillin, streptomycin, and 0.05% trypsin were provided by Mediatech (Washington, DC, USA). Fetal bovine serum (FBS) was obtained from Atlas Biologicals, Fort Collins, CO. While anti-SARS-CoV-2 spike S1 antibody was supplied by BioVision (Milpitas, CA, USA), anti-hACE2 antibody was provided by R&D Systems (Minneapolis, MN, USA). ACE2-interacting domain of SARS-CoV-2 (AIDS) peptides (>98% pure) were synthesized in GenScript (Piscataway, NJ, USA):Wild type (wt) AIDS: TNGVGYMutated (m) AIDS: TGGVGD)

Underlines indicate positions of mutations.

### 2.2. Cell Culture

A549 (human lung carcinoma; KRAS mut; EGFR wt) non-small cell lung cancer (NSCLC) cells were maintained at 37 °C and 5% CO_2_ in F12K media, supplemented with 10% FBS, 100 U/mL penicillin, and 100 µg/mL streptomycin. Once cells reached 80% confluence, these were passaged. Cells were washed with phosphate-buffered solution (PBS) and treated with 0.25% trypsin. Cells were suspended in F12K culture medium and seeded into T75 flasks.

Same procedure was utilized for two other cell lines purchased from ATCC:H1299 (human NSCLC, p53 negative; Catalog# CRL-5803)H358 (human NSCLC, KRAS mutant; Catalog# CRL-5807)

These cells were cultured in RPMI-1640 containing 10% FBS, 100 U/mL penicillin, and 100 µg/mL streptomycin. Cells from logarithmic phase were used for experiments.

### 2.3. Assessment of Viability in Cell Lines

#### 2.3.1. MTT Assay

The viability of cells was evaluated using the 3-(4,5-dimethylthiazol-2-yl)-2,5-diphenyl tetrazolium bromide (MTT) method with an in vitro toxicology assay kit from Sigma as described earlier [16,17]. It was used to measure mitochondrial activity. Cells were seeded in 24-well plates with 500 µL of F12K medium for 24 h followed by switching to serum-free medium. After 24 h of treatment just before adding MTT, 100 µL supernant was removed to be used for LDH assay (Figure 1). MTT was added to each well for 2 h according to the protocol outlined by the manufacturer. After removing the supernatant, formazan crystals were dissolved by adding equal volume of solution. At the end of the treatment period, 300 μL of culture medium was removed from each well and 20 μL of MTT solution (5 mg/mL) was added and incubated for 30 min. After distribution to a 96-well plate, absorbance was measured at 595 nm with the Thermo-Fisher Multiskan MCC plate reader (Fisher).

#### 2.3.2. Lactate Dehydrogenase Measurement

The activity of lactate dehydrogenase (LDH) was measured using a lactate dehydrogenase activity assay kit (Sigma) as described earlier [16,17,18]. A volume from the MTT assay was used and plated in a 96-well plate. An LDH master mix was prepared and added to each well. The reaction was carried out at room temperature in the dark. The resultant absorbance was measured at 450 nm with the Thermo-Fisher Multiskan MCC plate reader (Fisher).

### 2.4. Fragment end Labeling DNA

Fragmented DNA was detected in situ by the terminal deoxynucleotide transferase (TdT)-mediated binding of 3′OH ends of DNA fragments generated in response to apoptotic signals, using a commercially available kit (TdT FragEL DNA Detection Kit) from Millipore Sigma (Burlington, MA, USA) as described earlier [16,17,18]. Coverslips containing A549 lung adenocarcinoma cells cultured to 70–80% confluence were fixed with chilled methanol (Fisher Scientific, Waltham, MA, USA) for an hour, followed by two brief rinses with sterile PBS. Cover slips were treated with 20 mg/mL proteinase K for 5 min at room temperature and washed in PBS before TdT staining. Samples were equilibrated for 30 min in 1xTdT buffer and washed with PBS prior to terminal deoxynucleotidyl transferase and DAPI (1:10,000, Millipore Sigma, Burlington, MA, USA) staining. After mounting coverslips and drying overnight, slides were visualized under the Olympus BX41 fluorescent microscope equipped with a Hamamatsu ORCA-03G camera.

### 2.5. Immunostaining

Immunocytochemistry was performed by plating coverslips containing A549 cells cultured to 70–80% confluence as described before [19,20,21]. The cells were fixed with chilled Methanol (Fisher Scientific, Waltham, MA, USA) for one hour, followed by rinses with filtered PBS. Samples were blocked with 2% BSA (Thermo Fisher, Waltham, MA, USA) in PBS containing Tween 20 (Millipore Sigma, Burlington, MA, USA) and Triton X-100 (Millipore Sigma, Burlington, MA, USA) for 30 min and incubated at room temperature on a shaker. The primary antibodies used included: IFN-γ (1:100; Thermo Fisher, Waltham, MA, USA) incubated for 2 h on a shaker. After multiple washes in filtered PBS, coverslips were incubated with Cy5- labeled secondary antibody (1:200; Jackson ImmunoResearch, West Grove, PA, USA) for 1 h. After four washes in PBS, cells were incubated for 5 min in 4′,6′- diamindino-2- phenylindole (DAPI, 1:10,000; Millipore Sigma, Burlington, MA, USA). The coverslips were mounted and dried overnight then observed the Olympus BX41 fluorescent microscope equipped with a Hamamatsu ORCA-03G camera.

### 2.6. Annexin V and PI Flow Cytometry 

Single cell suspensions isolated from A549 cells were stained using the dead cell apoptosis kit with Annexin V for flow cytometry (Thermo Fisher, Waltham, MA, USA) according to manufacturer instructions as described before [22]. Cells were washed with Annexin V buffer and stained with Annexin V and PI (propidium iodide). Flow cytometry analyses were performed using the FACS Canto II Flow cytometer (BD Biosciences) and analyzed using FlowJo Software (v10). Only Annexin V, PI, and unstained cells served as control.

### 2.7. Immunoblotting

Western Blotting was conducted as described before [23,24]. Briefly, cells were harvested and lysed with lysis buffer containing 150 mM NaCl, 50 mM Tris (pH 8.0), 1% Triton-X, 0.1% SDS, 0.5% Na-Deoxycholate, and protease and phosphatase inhibitor cocktail to extract the total protein. The cells were transferred to microcentrifuge tubes and spun into a pellet. The supernatant was collected and analyzed for protein concentration via the Bradford method (Bio-Rad, Hercules, CA, USA). SDS sample buffer was added to 80–100 mg total protein and boiled for 10 min. Denatured samples were electrophoresed on Novex 15% Bis-Tris gels (Thermo Fisher, Waltham, MA, USA) and proteins transferred onto a nitrocellulose membrane (Bio-Rad, Hercules, CA, USA) using the BioRad Wet transfer. The membrane was washed for 10 min in PBS containing 0.1% Tween 20 (PBST) and blocked for 1 h in Intercept blocking buffer (Li-COR, Lincoln, NE). Next, membranes were incubated overnight at 4 °C under shaking conditions with the following primary antibodies at specified dilutions:Caspase-3 (Vendor: Santa-Cruz, Dallas, TX; Dilution: 1:200)Cleaved caspase-3 (Vendor: Cell Signaling, Danvers, MA; Dilution: 1:1000)p53 (Vendor: Santa-Cruz, Dallas, TX; Dilution: 1:200)Bcl2 (Vendor: Santa-Cruz, Dallas, TX; Dilution: 1:200)Bad (Vendor: Santa-Cruz, Dallas, TX; Dilution: 1:200)β-actin (Vendor: Abcam, Dallas, TX; Dilution: 1:10,000)

Actin was run as a loading control. The next day, membranes were washed in PBST for 30 min, and incubated with secondary antibodies (Vendor: Li-COR, Lincoln, NE; Dilution: 1:10,000) for 1 h at room temperature, washed in PBST for 30 min and visualized under the Odyssey Infrared Imaging System (Li-COR, Lincoln, NE, USA). Band intensities were quantified using Image J software.

### 2.8. Animals and Experimental Design: Intoxication of A/J Mice with NNK

Mice were maintained and experiments conducted in accordance with the National Institute of Health guidelines and approved by the Rush University Medical Center Institutional Animal Care and Use Committee (protocol # 22-007). Female A/J mice (6–8 week old) were obtained from Jackson Lab (Bar Harbor, ME, USA). NNK (sc-209854) was solubilized in saline and mice received NNK at a dose of 50 mg/kg body weight/week for two weeks via intraperitoneal injections. The protocol was adapted from [15,25] where the negative control mice received equal volume of saline (vehicle control).

#### 2.8.1. Treatment of NNK-Intoxicated Mice with Recombinant SARS-CoV-2 Spike S1 Protein

After 22 weeks of NNK intoxication, mice were treated with recombinant SARS-CoV-2 spike S1 intranasally at a dose of 50 ng/mouse/every other day. Recombinant spike S1 was dissolved in 4 μL normal saline, as described earlier [12,13] and mice were held in the supine position and 2 μL volume was delivered into each nostril using a pipet man and control mice received only normal saline.

#### 2.8.2. Tumor Histology

After 26 weeks of NNK intoxication, mice were euthanized with CO_2_. Tumors on the surface of the lungs were counted by a person blinded to the treatment regimens followed by taking picture of the whole lungs. Then, mice underwent transcardial perfusion as described before [26,27,28]. Lungs were excised, collected and processed for histological studies. Hematoxylin- eosin (HE) staining was performed from 5 μm paraffin embedded sections to study the morphology as described in [15]. The tumor area was analyzed by Image J, and ten images from 40× fields were chosen from each group.

### 2.9. Statistical Analysis

Statistical analyses were performed using Graphpad Prism 8 (GraphPad Software, Inc., La Jolla, CA, USA). Statistical differences between means were calculated by *t*-test (two-tailed). Variance between multiple means were conducted via one-way ANOVA, followed by Tukey’s post hoc tests. The criteria for statistical significance was *p* < 0.05. Values are expressed as means + SD of at least three independent experiments.

## 3. Results

### 3.1. Recombinant SARS-CoV-2 Spike S1 Treatment Induces Apoptosis and Death in Human A549 Lung Cancer Cells

It is commonly known that acquired resistance toward apoptosis and cell death is a hallmark of possibly all types of cancer [29]. Accordingly, pro-apoptotic approaches are beneficial for cancer as these strategies aim to trigger apoptosis in tumor cells [30]. To understand whether the SARS-CoV-2 spike S1 can break that resistance to cause cell death in lung cancer cells, human A549 NSCLC cells were challenged with recombinant spike S1.

We found dose-dependent increase in LDH release (Figure 2A) and decrease in MTT (Figure 2B) metabolism in A549 cells by recombinant spike S1. To confirm our observations, we performed dual FACS analysis with PI and annexin V (Figure 2C,D) and found that treatment with recombinant spike S1 significantly increased the level of annexin V-positive apoptotic cells in A549 cells.

To confirm the apoptosis further, we performed TUNEL staining and found increase in TUNEL-positive cells by spike S1 treatment (Figure 3A).

This was corroborated by counting of TUNEL-positive cells (Figure 3B). Several molecules are known to orchestrate apoptosis. For example, the caspases, specific cysteine proteases, play an integral role in the initiation and execution of apoptosis [30]. Similarly, the Bcl-2 antagonist of cell death (BAD) protein is known to regulate apoptosis by binding to anti-apoptotic members of the same family [31].

Immunoblot analysis of spike S1-treated and untreated A549 cells followed by quantification revealed increase in BAD and cleaved caspase-3 in spike S1-treated A549 cells as compared to control cells (Figure 3C–F). On the other hand, we observed a decrease in Bcl-2, an anti-apoptotic associated protein, upon spike S1 treatment (Figure 3G,H). Overall these results indicate that recombinant SARS-CoV-2 spike S1 treatment is capable of inducing apoptosis and cell death in human A549 NSCLC cells.

### 3.2. Recombinant SARS-CoV-2 Spike S1 Induces Death of Human A549 Lung Cancer Cells via Its Interaction with ACE2 Receptor

At first, to understand that the cell death induced by recombinant spike S1 is actually caused by spike S1 protein, not any contaminant present with the reagent, we used neutralizing antibodies against spike S1. Suppression of recombinant spike S1-induced cell death in A549 cells by neutralizing antibodies against spike S1, but not control IgG, suggests that cell death is in fact caused by spike S1 (Figure 4A,B). Studies have shown that the ACE-2 receptor expressed on the cell surface of the lung, heart, and kidneys [32] functions as a cellular receptor for SARS-CoV-2 spike S1 to enable viral entry into target cells [33,34]. Our immunostaining results also show the presence of ACE2 on the surface of A549 cells (Figure 4C).

Inhibition of recombinant spike S1-induced increase in LDH release (Figure 4D) and decrease in MTT metabolism (Figure 4E) in A549 cells by neutralizing antibodies against ACE2, but not control IgG (Figure 4D,E), suggests that spike S1 requires the involvement of ACE2 to induce cell death in A549 cells.

Recently we have described a different approach to dissociate the interaction between ACE2 and spike S1 of SARS-CoV-2. We have engineered a peptide corresponding to the ACE2-interacting domain of SARS-CoV-2 (AIDS) that inhibits the interaction between SARS-CoV-2 spike S1 and ACE2 [13]. Therefore, here we examined whether SARS-CoV-2 spike S1-intervened death of A549 cells was modulated by AIDS peptides. The wtAIDS, but not mAIDS, peptide inhibited spike S1-mediated increase in LDH release (Figure 5A) and decrease in MTT metabolism (Figure 5B) in A549 cells.

TUNEL staining also showed that wtAIDS, but not mAIDS, peptide was capable of neutralizing spike S1-mediated apoptosis of A549 cells (Figure 5C,D). Together, these results indicate the importance of spike S1protein binding to the ACE2 receptor to instigate an apoptotic response in human A549 lung cancer cells.

### 3.3. Recombinant SARS-CoV-2 Spike S1 Treatment Leads to Death of Human H1299 and H358 Lung Cancer Cells

Next, we wanted to investigate whether spike S1-induced death is specific for only A549 cells or it is seen in other human lung cancer cells as well. We used two more human NSCLC cell lines—H1299 and H358. Similar to A549 cells, spike S1 treatment dose-dependently increased LDH release in both H1299 (Figure 6A) and H358 (Figure 6B) cells. Accordingly, we also observed decrease in MTT metabolism in H1299 (Figure 6C) and H358 (Figure 6D) cells upon spike S1 treatment. Our results concluded that the stimulation of a death response is not specific to only A549 cancer cells, but other types of lung cancer as well.

### 3.4. Treatment with Recombinant SARS-CoV-2 Spike S1 Protein Causes Tumor Regression in NNK-Induced A/J Mice

Cell culture is a significantly different environment from that of a solid tumor. Therefore, to translate the cell culture finding to a clinically relevant setting, we used the A/J mice lung cancer model induced with 4-(methylnitrosamino)-1-(3-pyridyl)-1-butanone (NNK), a potent lung carcinogen [15]. This inducible animal model is a reliable method to recreate an authentic lung tumor environment. Lung carcinoma was induced in 5–6 week old female A/J mice with intraperitoneal injection of NNK (Figure 7A,B). However, our results showed that intranasal administration of recombinant SARS-CoV-2 spike S1 protein was capable of decreasing tumors in the lung of NNK-intoxicated A/J mice (Figure 7B). On the other hand, intranasal administration of normal saline had no such effect (Figure 7B). Analysis of lung histology demonstrated significant regression of lung tumors in spike S1-treated NNK-insulted mice as compared to control untreated NNK-challenged mice (Figure 7C). This was corroborated by quantification of tumor area (Figure 7D) and numbers (Figure 7E). These results indicate that spike S1 treatment is capable of causing tumor regression in NNK-challenged mice.

### 3.5. Intranasal Administration of Recombinant SARS-CoV-2 Spike S1 Protein Induces Apoptosis in Lung Tumor of NNK-Intoxicated A/J Mice

By evading apoptosis, an important hallmark of cancer [35], cancer cells stave off the apoptosis physiological process of cell death. We have seen that spike S1 treatment is capable of inducing apoptosis in A549 NSCLC cells (Figure 3).

Since intranasal administration of spike S1 led to the inhibition of lung tumor in NNK-intoxicated A/J mice (Figure 7), here, by using TUNEL, we examined whether spike S1 treatment could induce apoptosis in vivo in the lungs of NNK-challenged A/J mice. Although we hardly found the presence of TUNEL-positive cells in untreated NNK-insulted mice, the population of TUNEL-positive cells increased dramatically in the tumors of spike S1-treated NNK-confronted mice (Figure 8A). However, intranasal saline treatment remained unable to stimulate apoptosis (Figure 8A), indicating the specificity of the effect. These results were validated by counting of TUNEL-positive cells (Figure 8B). These results suggest that similar to that observed in human lung cancer cells, intranasal spike S1 treatment is capable of inducing apoptosis in vivo in lung tumors.

## 4. Discussion

NSCLC is an uncompromising type of lung cancer associated with limited treatment options. Drugs that inhibit tyrosine kinases EGFR (erlotinib, gefitinib, and afatinib) or ALK (crizotinib and ceritinib) have been approved for the treatment of NSCLC harbouring genetic modifications in the genes encoding these proteins [36,37,38]. Although these drugs are associated with a median progression-free survival of 9–14 months as compared to 5–7 months for platinum-based chemotherapy [39,40,41], improvements in overall survival have not been reported. With these and other therapeutic options, the median survival for NSCLC is 15.8 months and an overall 5-year survival rate for lung cancer patients is 22% [42]. As a result, lung cancer accounts for 25% of all cancer related deaths annually. Since NSCLC is chemotherapy sensitive it is the standard of care especially when surgery is not an option. Currently, multiple immunotherapies (e.g., modulators of PD-1/PD-L1, upregulators of tumor-associated macrophages, etc.) are also being tested for NSCLC in combination with various drugs [43,44,45].

Our studies presented here for the first time demonstrated that recombinant SARS-CoV-2 spike S1 protein stimulated cell death in human NSCLC cells. Our conclusion is dependent on the following observations: First, treatment with recombinant SARS-CoV-2 spike S1 protein prompted cell death in human A549, H1299 and H358 NSCLC cells. Second, FACS staining with PI and annexin V uncovered an increase in early apoptotic and late apoptotic cells in human A549 NSCLC cells upon treatment with spike S1 protein.

Third, the number of TUNEL positive cells was much higher in spike S1-treated A549 NSCLC cells as compared to control. Fourth, levels of apoptosis-related proteins such as BAD and cleaved caspase-3 were higher in spike S1-treated A549 NSCLC cells as compared to control. Fifth, we also found that cell survival-associated molecule Bcl-2 was downregulated in A549 NSCLC cells following spike S1 treatment. These results suggest that recombinant SARS-CoV-2 spike S1 protein may be considered to eradicate human NSCLC cells.

Many times therapeutic strategies generated from cultured cells are not translated to in vivo in tumor because of limited or no access of the drug to the target organ and associated tumor microenvironment. Therefore, here, we did not use the xenograft model, but NNK intoxicated mouse model where tumors form directly in the lung [46,47]. To meet animal noncompliance from the IACUC, these mice are generally sacrificed within 26–27 weeks of NNK intoxication. Usually, patients are treated with a drug or therapy after the diagnosis. Therefore, we started treatment with recombinant SARS-CoV-2 spike S1 from 22 weeks of NNK insult, which should be considered as a late stage of the disease. However, alternate day intranasal treatment of low dose of recombinant spike S1 for 4 weeks led to the decrease in both number and size of tumors in the lungs of NNK-insulted mice. TUNEL staining of lung sections also indicated that intranasal spike S1 treatment was capable of inducing apoptosis in lung tumors of NNK-intoxicated mice. These results suggest intranasal administration of SARS-CoV-2 spike S1 may have therapeutic implications in the regression of tumor in lung cancer patients.

Mechanisms by which SARS-CoV-2 spike S1 induces apoptosis are poorly understood. However, it is known that the prototype receptor of spike S1 is the ACE2 receptor, which is overexpressed in non-small cell lung cancer (NSCLC) [7,8]. We have also seen the presence of ACE2 in human A549 NSCLC cells. Attenuation of recombinant spike S1-mediated death of A549 cells by neutralizing antibodies against ACE2 indicates the involvement of ACE2 in the oncolytic effect of recombinant spike S1.

Recently we have seen that a peptide corresponding to the ACE2-interacting domain of SARS-CoV-2 (AIDS) inhibits the interaction between SARS-CoV-2 spike S1 and ACE2 [13]. Interestingly, wtAIDS, but not mAIDS, peptide inhibited recombinant spike S1-mediated apoptosis and cell death in A549 cells. Together, these results delineate that the interaction between spike S1 and ACE2 is required for spike S1-mediated death of A549 cells.

How does the interaction between ACE2 and spike S1 lead to cell death? Inflammation driven by proinflammatory cytokines plays an important role in the pathogenesis of many human disorders including COVID-19. Accordingly, several studies have shown that many COVID-19 patients suffer from cytokine storm [48]. Recently, we have described that recombinant SARS-CoV-2 spike S1 alone is capable of driving the expression of proinflammatory cytokines including TNFα in human A549 NSCLC cells [13]. Consistent to the involvement of NF-κB activation in the expression of different proinflammatory molecules [49,50], we have also seen that spike S1 intoxication stimulates the activation of NF-κB in cultured A549 cells and in vivo in the lungs [12,13,51]. Therefore, spike S1 can drive (NF-κB—proinflammatory cytokine)-mediated inflammatory immune response in A549 cells. Since several studies have reported that proinflammatory cytokines like TNFα is capable of inducing cancer cell death [52,53], it is possible that intranasal SARS-CoV-2 spike S1 can elicit the death of lung cancer cells via proinflammatory cytokines.

## 5. Conclusions

In summary, our studies demonstrate that recombinant SARS-CoV-2 spike S1 causes death of human NSCLC cells via its interaction with ACE2 and that intranasal administration of recombinant spike S1 leads to regression of cancer in vivo in the lung of NNK-intoxicated mouse model. Although the NNK-induced mouse model of lung cancer may not correctly replicate the in vivo condition of lung tumors in human lung cancer patients, our results highlight a new therapeutic avenue for late stage lung cancer patients with recombinant SARS-CoV-2 spike S1 protein.

## 6. Limitations

Since this putative therapeutic approach involves spike S1 and most of the available COVID-19 vaccines neutralize the function of SARS-CoV-2 spike S1, COVID-19 vaccines should reduce the efficacy of this putative virotherapy. In that case, whether lung cancer patients might be kept away from COVID-19 vaccines or not needs further investigation.

## Figures and Tables

**Figure 1 cancers-14-05648-f001:**
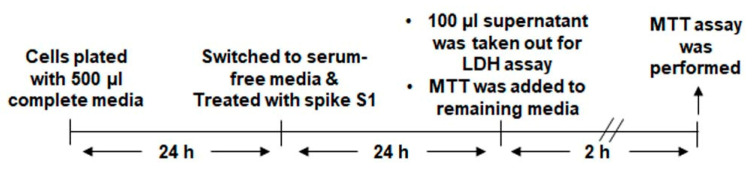
Time line for LDH and MTT assays.

**Figure 2 cancers-14-05648-f002:**
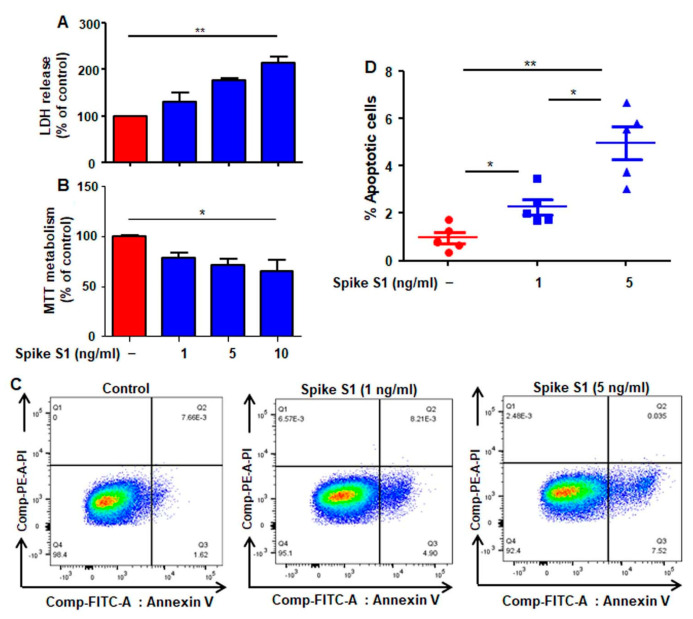
Effect of recombinant SARS-CoV-2 spike S1 on the survival of human A549 lung cancer cells. A549 cells were treated with different concentrations (1, 5, and 10 ng/mL) of recombinant spike S1 protein for 24 h under serum-free condition followed by monitoring cell death by LDH release (**A**) and MTT (**B**). FACS double staining with annexin V and propidium iodide (PI) was also performed (**C**). Quantitative analysis of percent apoptotic cells is presented (**D**). Values are presented as mean ± SD of three independent experiments. * *p* < 0.05; ** *p* < 0.01.

**Figure 3 cancers-14-05648-f003:**
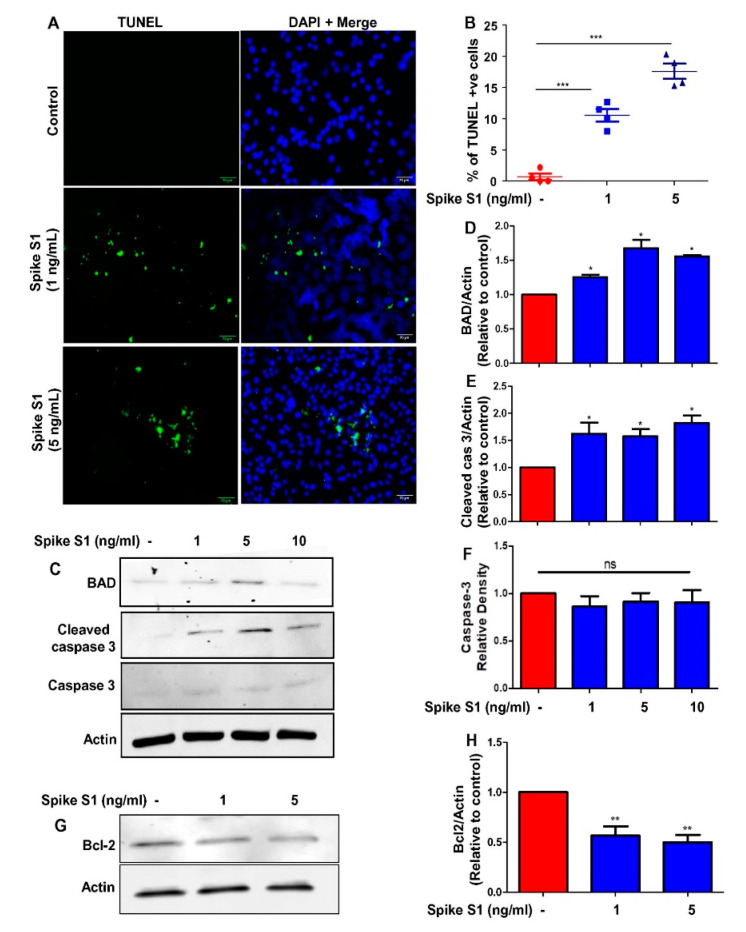
Recombinant SARS-CoV-2 spike S1 induces apoptosis in human A549 lung cancer cells. A549 cells were treated with different doses of spike S1 protein for 12 h under serum-free condition followed by monitoring apoptosis by TUNEL (**A**). TUNEL positive cells were counted in 10 varied images per group and plotted as percent of total cells (**B**). (**C**) Cells were immunoblotted for apoptosis-related molecules (BAD, caspase 3 and cleaved caspase 3). Actin was run as a loading control. For original blots, please see Appendix A. Bands were scanned and values ((**D**), BAD/Actin; (**E**) cleaved caspase 3/Actin; (**F**) caspase 3/Actin) presented as relative to control. (**G**) Cells were immunoblotted for survival-related molecule (Bcl_2_). Actin was run as a loading control. (**H**) Bands were scanned and values (Bcl_2_/Actin) presented as relative to control. Results are mean ± SD of three different experiments. * *p* < 0.05; ** *p* < 0.01; *** *p* < 0.001; ns, not significant. Scale bar = 22 µm.

**Figure 4 cancers-14-05648-f004:**
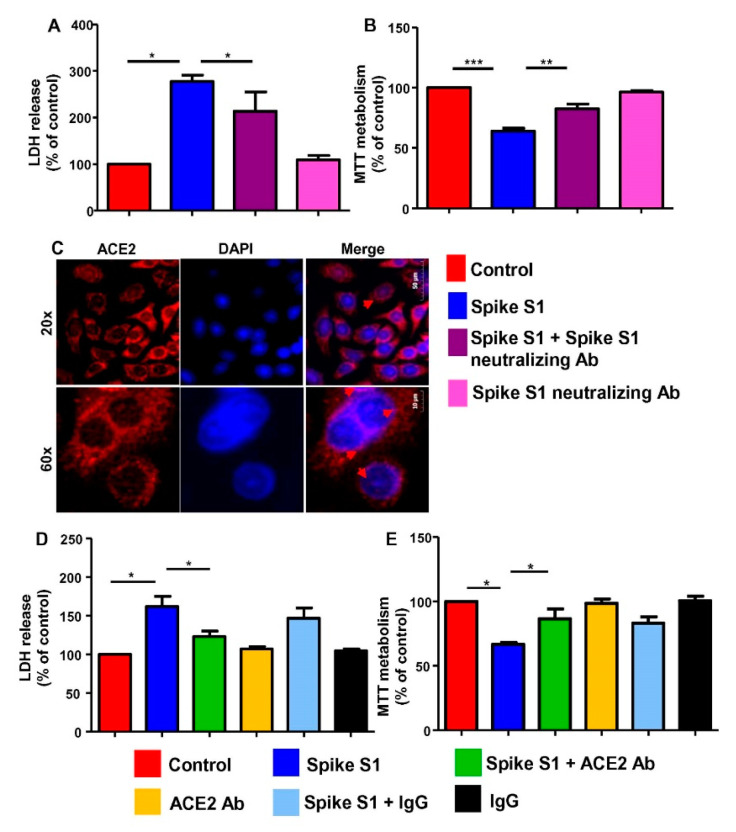
Spike S1-mediated death of human A549 lung cancer cells depends on ACE2 receptor. A549 cells were treated with 5 ng/mL spike S1 protein in the presence or absence of neutralizing antibodies against spike S1 (0.5 µg/mL) under serum-free condition. After 24 h, cell viability was monitored by LDH release (**A**) and MTT (**B**). Control A549 cells were immunostained for ACE2 (**C**). DAPI was used to stain nuclei. Cells were treated with 5 ng/mL spike S1 protein in the presence or absence of neutralizing antibodies against ACE2 (0.5 µg/mL) under serum-free condition. After 24 h, cell viability was monitored by LDH release (**D**) and MTT (**E**). Results are mean ± SD of three different experiments. * *p* < 0.05; ** *p* < 0.01; *** *p*< 0.001. Scale bar = 10 µm.

**Figure 5 cancers-14-05648-f005:**
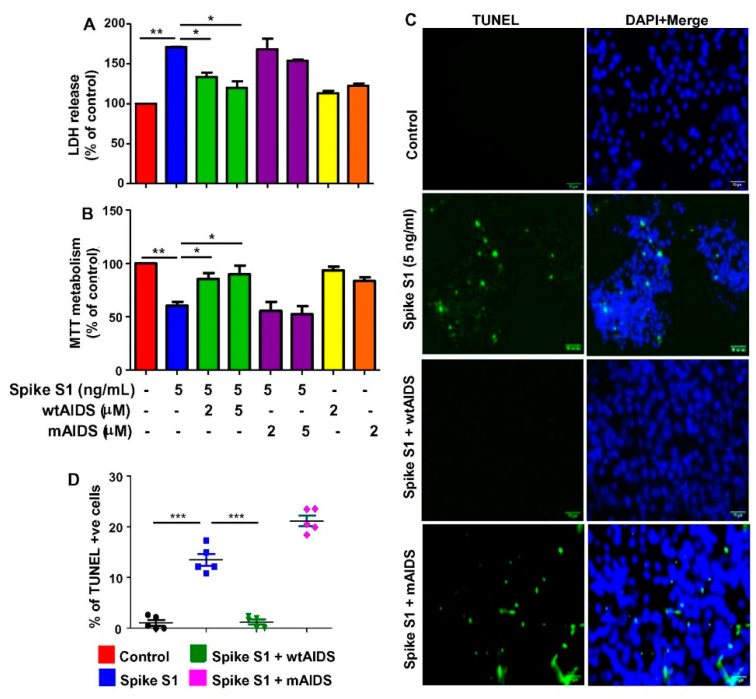
Selective disruption of ACE2-to-spike S1 interaction reduces spike S1-induced death in human A549 lung cancer cells. A549 cells were treated with 5 ng/mL spike S1 protein in the presence or absence of different concentrations of wtAIDS and mAIDS peptides under serum-free condition. After 24 h, cell viability was monitored by LDH release (**A**) and MTT (**B**). After 12 h of treatment, apoptosis was monitored by TUNEL (**C**). TUNEL positive cells were counted in 10 varied images per group and plotted as percent of total cells (**D**). Results are mean ± SD of three different experiments. * *p* < 0.05; ** *p* < 0.01; *** *p*< 0.001. Scale bar = 22 µm.

**Figure 6 cancers-14-05648-f006:**
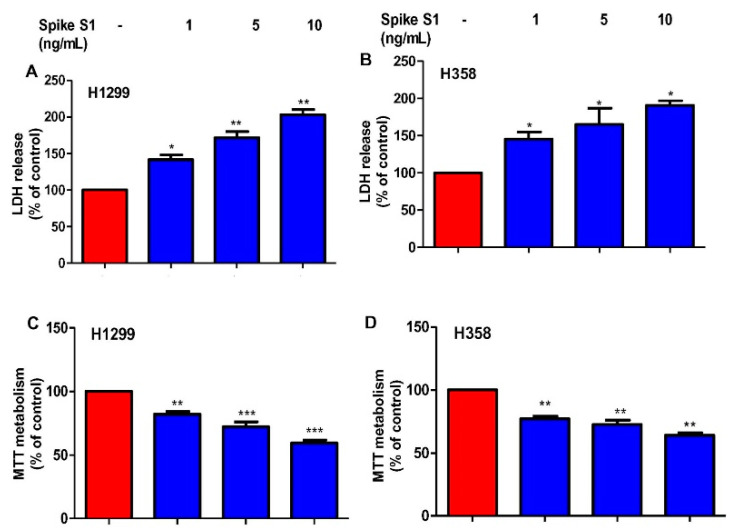
Effect of recombinant SARS-CoV-2 spike S1 on the survival of human H1299 and H358 lung cancer cells. H1299 (**A**,**C**) and H358 (**B**,**D**) cells were treated with spike S1 protein for 24 h under serum-free condition followed by monitoring cell death by LDH release (**A**,**B**) and MTT (**C**,**D**). Results are mean + SD of three different experiments. * *p* < 0.05; ** *p* < 0.01; *** *p*< 0.001.

**Figure 7 cancers-14-05648-f007:**
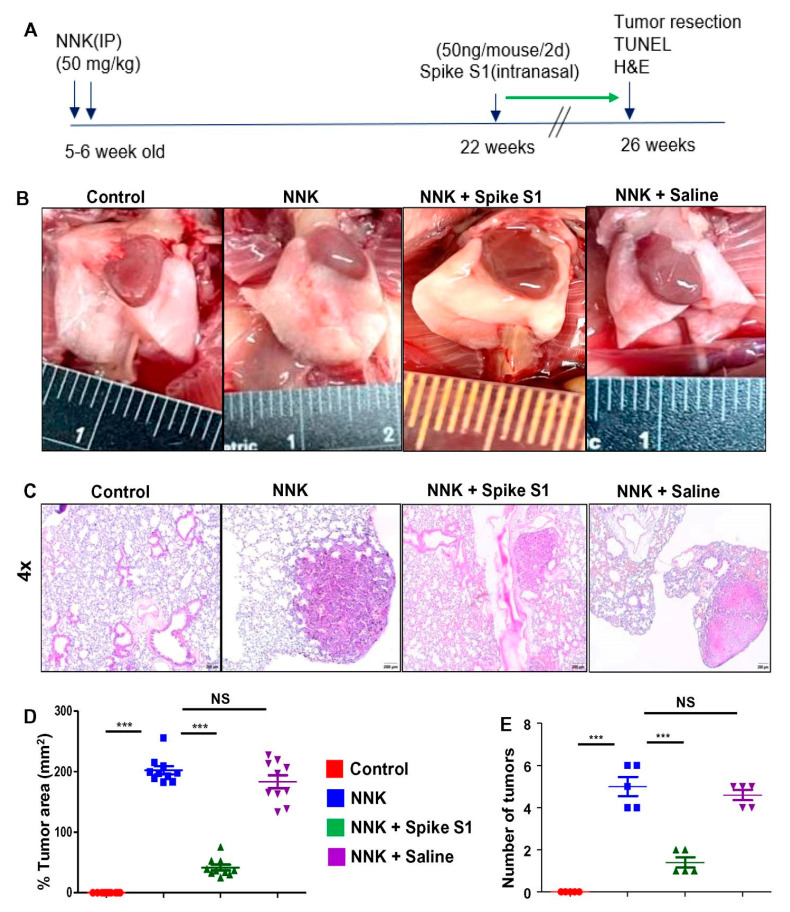
Intranasal administration of recombinant SARS-CoV-2 spike S1 causes regression of lung tumor in NNK-insulted female A/J mice. The experimental design is illustrated for NNK-induced lung cancer in A/J mice (**A**). Briefly, female A/J mice (5–6 week old) received two intraperitoneal (i.p.) injections of NNK (50 mg/kg body weight) one week apart. Tumor development was analyzed after 26 weeks of NNK intoxication. Mice were treated with spike S1 (50 ng/mouse/2 d) intranasally on alternate days starting from 22 weeks of NNK insult for 4 weeks followed by sacrificing mice on 26 weeks. Representative lung appearance in different groups of mice (**B**). Lung sections were stained for H&E (**C**). The histological tumor area was quantified in a 4x field as a percent of control (**D**). The number of lung lesions is shown in different groups of mice (**E**). Results are mean ± SD of 5 mice per group. *** *p*< 0.001; NS, not significant. Scale bar = 20 µm.

**Figure 8 cancers-14-05648-f008:**
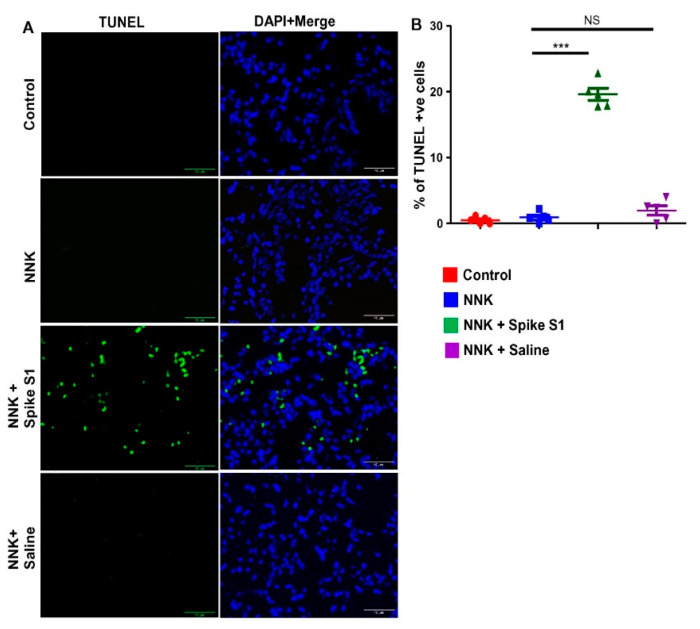
Intranasal administration of recombinant SARS-CoV-2 spike S1 induces apoptosis in lung tumors of NNK-insulted female A/J mice. Female A/J mice (5–6 week old) received two intraperitoneal (i.p.) injections of NNK (50 mg/kg body weight) one week apart. Mice were treated with spike S1 (50 ng/mouse/2 d) intranasally on alternate days starting from 22 weeks of NNK insult for 4 weeks followed by sacrificing mice on 26 weeks. Tumor tissue sections were labeled for TUNEL (**A**) followed by counting of TUNEL-positive cells in two sections (two images per section) of each of five mice per group (**B**). Results are mean + SD of 5 mice per group. *** *p* < 0.001; NS, not significant. Scale bar = 40 µm.

## Data Availability

All data are present in this manuscript.

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
