# Peer review of "Regression of Lung Cancer in Mice by Intranasal Administration of SARS-CoV-2 Spike S1"

_cancers, 2022, doi:10.3390/cancers14225648_

Round 1

Reviewer 1 Report

The manuscript “Regression of lung tumor in mice by intranasal SARS-CoV-2 spike S1” by Sheinin et al. addresses an interesting function of the spike protein S1 that has evolved from studies on the entry and infectivity of the SARS-CoV-2 virus. Multiple studies have shown that SARS-CoV-2 binds to its receptor ACE2 via spike protein S1 to enter the cells. The authors found an interesting and important observation, wherein; recombinant SARS-CoV2 spike S! employed its interaction with ACE2 to induce death of human lung cancer cells. Furthermore, intranasal administration of recombinant spike S1 led to regression of tumor in vivo in the lung of NNK- intoxicated mice; suggesting that intranasal administration of SARS-CoV2 spike S1 may be beneficial to lung cancer patients. 

The authors demonstrate that SARS –CoV-2 spike S1 is important in inducing apoptotic cell death in lung cancer. The ACE2- interaction domain of spike S1 is necessary for the induction of cell death. With an animal model of NSCLC, using chemically induced tumor in lungs of mice; a low dose intranasal treatment with recombinant SARS-CoV-2 spike S1 showed tumor regression in the lungs induced by apoptosis.  Relevant in vitro human NSCLC cell lines have been used along with a murine in vivo lung cancer model. The experiments showing apoptotic cell death are comprehensive and well designed.

The manuscript adds timely to the mechanistic knowledge. The work is original and has the potential for understanding the molecular biology of human disease. The experiments are well designed, the methods are rigorous, and the results are credible and impressive. The discussion is also appropriate. This paper represents a major breakthrough in lung cancer research.

Author Response

Reviewer 1:

Comment: The manuscript “Regression of lung tumor in mice by intranasal SARS-CoV-2 spike S1” by Sheinin et al. addresses an interesting function of the spike protein S1 that has evolved from studies on the entry and infectivity of the SARS-CoV-2 virus. Multiple studies have shown that SARS-CoV-2 binds to its receptor ACE2 via spike protein S1 to enter the cells. The authors found an interesting and important observation, wherein; recombinant SARS-CoV2 spike S1 employed its interaction with ACE2 to induce death of human lung cancer cells. Furthermore, intranasal administration of recombinant spike S1 led to regression of tumor in vivo in the lung of NNK- intoxicated mice; suggesting that intranasal administration of SARS-CoV2 spike S1 may be beneficial to lung cancer patients. 

The authors demonstrate that SARS –CoV-2 spike S1 is important in inducing apoptotic cell death in lung cancer. The ACE2- interaction domain of spike S1 is necessary for the induction of cell death. With an animal model of NSCLC, using chemically induced tumor in lungs of mice; a low dose intranasal treatment with recombinant SARS-CoV-2 spike S1 showed tumor regression in the lungs induced by apoptosis.  Relevant in vitro human NSCLC cell lines have been used along with a murine in vivo lung cancer model. The experiments showing apoptotic cell death are comprehensive and well designed.

The manuscript adds timely to the mechanistic knowledge. The work is original and has the potential for understanding the molecular biology of human disease. The experiments are well designed, the methods are rigorous, and the results are credible and impressive. The discussion is also appropriate. This paper represents a major breakthrough in lung cancer research.

Response: Thank you very much.

Reviewer 2 Report

In this preclinical study the authors investigate the effect of recombinant SARS-CoV-2 spike S1 protein on NSCLC. The rationale behind this study emerges from observations on the regulatory role of ACE-2 on cancer growth. The authors hypothesized that employing recombinant SARS-CoV-2 S1 protein could regulate ACE2 expression and subsequently lung cancer cell growth.

The strong element of this study relies on the methodology, as the investigators used several controls (assay-to-assay and subject-to-subject controlling) to demonstrate their reproducibility on two different levels. Firstly, they used well annotated lung cancer cell lines to test their hypothesis. Then they controlled their results in vivo in non-human subjects (NNK intoxicated mice).

The main disadvantage of this paper lies on the manuscript presentation, mainly on methods.

As so, I believe that this paper should be accepted for publication under one major and several minor revisions.

Two general comments for the authors are:

i)                    Please try to make distinction of what should be said in methods, what in results, and what in discussion

ii)                   There are some odd statements regarding NSCLC, throughout introduction and discussion. It would be useful to visit ASCO or NCCN guidelines for NSCLC management.

Please consider the following comments for manuscript improvement:

Major comments

The authors went through a detailed tech description of their assays in methods section. Though not of great interest for average reader without lab experience, that would be acceptable if there was not a gap of the actual pipeline description in the methods.

The authors have transferred the actual methods description, along with some introductory mentions, in the result section.

Please see some typical examples:

Line 199. “To understand whether the SARS-CoV-2 spike S1 can break that resistance to cause cell death in lung cancer cells, human A549 NSCLC 200 cells were incubated with different concentrations (1, 5, and 10 ng/mL) of recombinant spike S1 under serum-free conditions followed by measuring cell survival by LDH release and MTT assay.”

Line 241. “At first, to understand that the cell death induced by spike S1 is actually caused by spike S1, not any contaminant present with the reagent, we used neutralizing antibodies  against spike S1.”

Line 245. “Studies have shown that the ACE-2 receptor is expressed on the cell surface of the lung, heart, and kidneys [30]. ACE2 functions as a cellular  receptor for spike S1 protein to enable viral entry into target cells. Our immunostaining results show the presence of ACE2 in A549 cells (Fig. 3C). Therefore, next, we used neutralizing antibodies against ACE2 and found inhibition of recombinant spike S1-induced death of A549 cells by neutralizing antibodies against ACE2, but not control IgG (Fig. 3D-E), suggesting that spike S1 requires the involvement of ACE2 to induce cell death in A549  cells.”

I believe that methods (and consequently results) need major restructuring.

My suggestion for the authors is:

i)                    Provide a supplement for detailed assays description and remove them from methods

ii)                   Go through detailed description of your workflow in methods (for example use chapters explaining why, what, and how you performed each step: “assessment of viability in cell lines” à “assessment of SARS-CoV-2 S1 coupling with ACE-2” à “Cell line control” à “In vivo assessment of SARS-CoV-2 S” etc.)

iii)              Remove the above-mentioned description from results section

Minor comments

Title

1)      Please be more specific. What kind of lung tumor? Also, please prefer the term cancer, as tumor is a more general term.

2)      Please state clearly that regression is associated with a drug administration (I would suggest using the words “administration” and “recombinant” spike S1)

Simple summary

3)      Line 11, “Although there are some therapies against lung cancer, new effective treatment options are badly needed.”

Vague statement. Undermines the huge developments in lung cancer management over the last decade.

4)      Please try to be more specific regarding lung cancer definition throughout the text. The study was conducted specifically for non-small cell lung cancer. I doubt if the results are reproducible in small cell or other endocrine tumors.

Abstract

5)      Please provide a structured abstract.

6)      Line 20. Please rephrase “…inducing death in lung  cancer”. This statement seems like the drug induces death in individuals with lung cancer. I believe you meant to say apoptosis in lung cancer cells.

7)      I believe selection of keywords is not the best. Expanding and modifying some of the keywords will help both the authors and the journal to future citations.

Introduction

8)      Line 40, “Ironically, among all cancers, lung cancer is by far the leading cause of cancer death in which more than 130,000 people 41 die each year in USA”

Why ironically? Please explain or rephrase.

9)      Paragraph 2. The authors try to explain the methods and their results. Methods/results should not be in the introductory section. The paragraph needs also some work on syntax, as it is difficult to follow the sentences.

Please restructure (or remove) this paragraph, without details on methods and results.

I miss a paragraph on introduction about what is known on the topic, what other similar studies have shown (for example there are several studies on ACE2 inhibitors in lung cancer), and a brief description of what you are going to tell us in the paper.

Methods

10)   Reagents: Please use other verbs (instead of purchase, buy) which do not reflect transaction (eg provide, supply)

The rest as described in major comments.

Results

Again the same issue as mentioned before. There is a synthesis of introductory statements, methods description, and results given. Please try to clear that out.

11)   Figures

i)                    It would be better if you title y axis as “LDH” and “MTT metabolism” rather than “Cytotoxicity” and “viability”. You should explain in the text the association of LDH and MTT with cytotoxicity and cell viability

ii)                   Please give titles in all x-axes (in all bar charts figures of the paper)

iii)                 Please define different colors in all figures of the paper

iv)                 The results of analyses for statistically significant differences should be provided in the text or as supplementary material (besides figures)

v)                   Figure 5. You should use bars/lines to demonstrate the comparisons

12)   Be sure that every abbreviation is explained upon first mention (i.e. line 285, NSCLC)

Discussion

13)   Line 351. Median survival is not expressed in percentage. Please rephrase.

14)   Line 352-354. The statement that chemotherapy is the standard of care for NSCLC and that many immunotherapies are being tested is untrue. Please visit the ASCO or NCCN guidelines for NSCLC management. NSCLC includes combination of chemotherapy, immunotherapy, radiotherapy, targeted therapy, and surgery in the first line setting (and beyond) according to stage.

15)   Please describe limitations of this study

References

16)   Out of the 42 references, 21 are self-citations (50%). I understand that the authors may be experts on the field, however anything >20% is considered unacceptable in many journals. The authors should try to enrich their references with non-self publications (or reduce self-citations)

Author Response

Reviewer 2:

Comment: In this preclinical study the authors investigate the effect of recombinant SARS-CoV-2 spike S1 protein on NSCLC. The rationale behind this study emerges from observations on the regulatory role of ACE-2 on cancer growth. The authors hypothesized that employing recombinant SARS-CoV-2 S1 protein could regulate ACE2 expression and subsequently lung cancer cell growth.

The strong element of this study relies on the methodology, as the investigators used several controls (assay-to-assay and subject-to-subject controlling) to demonstrate their reproducibility on two different levels. Firstly, they used well annotated lung cancer cell lines to test their hypothesis. Then they controlled their results in vivo in non-human subjects (NNK intoxicated mice).

Response: Thank you.

Comment: The main disadvantage of this paper lies on the manuscript presentation, mainly on methods.

As so, I believe that this paper should be accepted for publication under one major and several minor revisions.

Two general comments for the authors are:

  1. i)                    Please try to make distinction of what should be said in methods, what in results, and what in discussion
  2. ii)                   There are some odd statements regarding NSCLC, throughout introduction and discussion. It would be useful to visit ASCO or NCCN guidelines for NSCLC management.

Response: We have taken care of all the comments. For easy tracking all changes are highlighted. Thank you.

Comment: Please consider the following comments for manuscript improvement:

Major comments

The authors went through a detailed tech description of their assays in methods section. Though not of great interest for average reader without lab experience, that would be acceptable if there was not a gap of the actual pipeline description in the methods.

The authors have transferred the actual methods description, along with some introductory mentions, in the result section.

Please see some typical examples:

Line 199. “To understand whether the SARS-CoV-2 spike S1 can break that resistance to cause cell death in lung cancer cells, human A549 NSCLC 200 cells were incubated with different concentrations (1, 5, and 10 ng/mL) of recombinant spike S1 under serum-free conditions followed by measuring cell survival by LDH release and MTT assay.”

Line 241. “At first, to understand that the cell death induced by spike S1 is actually caused by spike S1, not any contaminant present with the reagent, we used neutralizing antibodies  against spike S1.”

Line 245. “Studies have shown that the ACE-2 receptor is expressed on the cell surface of the lung, heart, and kidneys [30]. ACE2 functions as a cellular  receptor for spike S1 protein to enable viral entry into target cells. Our immunostaining results show the presence of ACE2 in A549 cells (Fig. 3C). Therefore, next, we used neutralizing antibodies against ACE2 and found inhibition of recombinant spike S1-induced death of A549 cells by neutralizing antibodies against ACE2, but not control IgG (Fig. 3D-E), suggesting that spike S1 requires the involvement of ACE2 to induce cell death in A549  cells.”

 Response: We have taken care of all these comments and modified accordingly. Thank you.

Comment: I believe that methods (and consequently results) need major restructuring.

My suggestion for the authors is:

  1. i)                    Provide a supplement for detailed assays description and remove them from methods
  2. ii)                   Go through detailed description of your workflow in methods (for example use chapters explaining why, what, and how you performed each step: “assessment of viability in cell lines” à “assessment of SARS-CoV-2 S1 coupling with ACE-2” à “Cell line control” à “In vivo assessment of SARS-CoV-2 S” etc.)

iii)              Remove the above-mentioned description from results section

Response: We have written the methods in detail and provided a timeline for survival assays. Thank you.

Minor comments

Comments: Title

1)      Please be more specific. What kind of lung tumor? Also, please prefer the term cancer, as tumor is a more general term.

2)      Please state clearly that regression is associated with a drug administration (I would suggest using the words “administration” and “recombinant” spike S1)

Simple summary

3)      Line 11, “Although there are some therapies against lung cancer, new effective treatment options are badly needed.”

Vague statement. Undermines the huge developments in lung cancer management over the last decade.

4)      Please try to be more specific regarding lung cancer definition throughout the text. The study was conducted specifically for non-small cell lung cancer. I doubt if the results are reproducible in small cell or other endocrine tumors.

Response: We have modified these as suggested. Thank you.

Comments: Abstract

5)      Please provide a structured abstract.

6)      Line 20. Please rephrase “…inducing death in lung  cancer”. This statement seems like the drug induces death in individuals with lung cancer. I believe you meant to say apoptosis in lung cancer cells.

7)      I believe selection of keywords is not the best. Expanding and modifying some of the keywords will help both the authors and the journal to future citations.

Response: We have taken care of these. Thank you.

Comments: Introduction

8)      Line 40, “Ironically, among all cancers, lung cancer is by far the leading cause of cancer death in which more than 130,000 people 41 die each year in USA”

Why ironically? Please explain or rephrase.

9)      Paragraph 2. The authors try to explain the methods and their results. Methods/results should not be in the introductory section. The paragraph needs also some work on syntax, as it is difficult to follow the sentences.

Please restructure (or remove) this paragraph, without details on methods and results.

I miss a paragraph on introduction about what is known on the topic, what other similar studies have shown (for example there are several studies on ACE2 inhibitors in lung cancer), and a brief description of what you are going to tell us in the paper.

Response: We have modified these accordingly. Thank you.

Comments: Methods

10)   Reagents: Please use other verbs (instead of purchase, buy) which do not reflect transaction (eg provide, supply)

The rest as described in major comments.

Response: We have modified these as suggested. Thank you.

Comments: Results

Again the same issue as mentioned before. There is a synthesis of introductory statements, methods description, and results given. Please try to clear that out.

11)   Figures

  1. i)                    It would be better if you title y axis as “LDH” and “MTT metabolism” rather than “Cytotoxicity” and “viability”. You should explain in the text the association of LDH and MTT with cytotoxicity and cell viability
  2. ii)                   Please give titles in all x-axes (in all bar charts figures of the paper)

iii)                 Please define different colors in all figures of the paper

  1. iv)                 The results of analyses for statistically significant differences should be provided in the text or as supplementary material (besides figures)
  2. v)                   Figure 5. You should use bars/lines to demonstrate the comparisons

12)   Be sure that every abbreviation is explained upon first mention (i.e. line 285, NSCLC)

Response: We have taken care of these and modified accordingly. Thank you.

Comments: Discussion

13)   Line 351. Median survival is not expressed in percentage. Please rephrase.

14)   Line 352-354. The statement that chemotherapy is the standard of care for NSCLC and that many immunotherapies are being tested is untrue. Please visit the ASCO or NCCN guidelines for NSCLC management. NSCLC includes combination of chemotherapy, immunotherapy, radiotherapy, targeted therapy, and surgery in the first line setting (and beyond) according to stage.

15)   Please describe limitations of this study

Response: We have taken care of these. Thank you.

Comments: References

16)   Out of the 42 references, 21 are self-citations (50%). I understand that the authors may be experts on the field, however anything >20% is considered unacceptable in many journals. The authors should try to enrich their references with non-self publications (or reduce self-citations)

Response: We have reduced self-citations. Thank you.